# MSW Management to Zero Waste: Challenges and Perspectives in Belarus

Alena Harbiankova [1,*] and Sławomir Kalinowski [2]

1   Independent Researcher, 02-414 Warsaw, Poland
2   Department of Rural Economics, Institute of Rural and Agricultural Development, Polish Academy of Sciences, ul. Nowy Swiat 72, 00-330 Warsaw, Poland
*   Correspondence: a.harbiankova@gmail.com

**Abstract:** Municipal solid waste management is one of the key challenges of environmental, economic and social importance. It is a global problem regardless of economic development level and political orientation, and also applies to a country such as Belarus. There is a lack of studies considering the local aspects of waste management in Belarus, therefore the research is actual. The study aims to formulate the recommendations on the conceptual modelling of the MSW management system in Belarus. The research methods are based on the complex systems approach. The study provides a review of the actual concepts on MSW management, including its general principles, hierarchy and methods, analysis of the current state of MSW management system in Belarus, and recommendations on modeling the MSW management system covering the local and regional aspects in Belarus. The suggestions on formulating the MSW management system involve organizational, economic, technical and informational bases as well as considering the local and regional specifics. The results indicate the following: (1) the current MSW management system in Belarus does not move the country towards a circular economy, requiring an update of the existing waste management strategy; (2) sustainable waste management demands an integrated approach in order to support waste recycling into a manufacturing resource within circular economy; (3) two main approaches to solve the problem of waste management in rural communities have been identified— an economic approach implies the minimization of waste generation while a technological approach comprises the development of mini-solutions for waste recycling at the places of waste generation. The research results contribute to the increased interest in the issue of MSW management in Belarus, and can be a useful tool for improving the planning strategies considering the local and regional context.

**Keywords:** zero waste; municipal solid waste; waste management; sustainable development; regional planning



## 1. Introduction

Waste management is a major challenge of environmental, economic and social importance all over the world. It is recognized as one of the sources of inefficiency in the waste management system, constituting climate change, growing plastic pollution, and food loss, which globally translates into 6.3 billion tons of plastic waste [1], 1.6 billion metric tons of carbon dioxide [2], and 931 million tons of food waste [3]. It is an interdisciplinary problem requiring sustainable solutions [4,5] without regard to national economic development [6,7], local specifics [8–10], and political orientation [11,12]. Globally, the most attention is focused on highly hazardous [13,14], radioactive [15,16] and toxic waste [17,18]. However, the issue of management of industrial and consumption waste, including municipal solid waste (MSW), is equally important [19,20]. A significant volume of waste comes from food losses and waste [21,22] resulting in various economic, environmental and social impacts [23,24] including a danger for the global food security [25]. The urgency of the problem is provided by the fact that MSW volume is constantly increasing, both in absolute

values and per capita [26], while inadequate MSW management causes both local and global impacts [27].

Direct waste impact on the environment involves climate change as well as pollution of air, water, and soil [28,29]. Total GHG emissions due to ineffective waste management by treatment and recycling contribute up to 5% of the overall GHG emissions [30,31], while waste has the potential to be a production resource and a clear GHG sink due to effective waste treatment [32]. MSW is considered as the dominant factor of environmental stress [33]: over 50% of the collected MSW is poorly managed as burned or landfilled, resulting in GHG emissions [34,35].

The social dimension of waste management involves reducing the direct and indirect effects on public health and life quality [36,37]. The significant damage is caused by air pollutants from waste-to-energy facilities [38,39]. Inefficient waste management causes pollution of water basins and drinking water sources and cultivation of agricultural crops in polluted soils [40–42]. The annual damage to the marine ecosystem caused by plastic is valued as USD 13 billion a year, while the total value of plastic production is estimated as USD 75 billion [43].

Economic dimensions of an ineffective waste management are related to resource losses at the stages of extraction, production, distribution, and consumption [44–46]. According to a UNEP report 17 per cent of global food production was wasted in 2019, meanwhile a significant rate of 61 per cent came from households [2]. Reducing food waste has both social and economic benefits [47], which in figures amount to feeding 214 millions hungry people and about USD 68 billion annually [48,49].

These negative effects due to inefficient MSW management in the environmental, economic and social aspects justify the need for intensification of research in this field and then the urgent need to take preventive measures. All the above also apply to a country such as Belarus. Nowadays, in the face of the global environmental crisis, this is of colossal importance when one thinks of achieving SDG 11 for Sustainable cities and communities and SDG 12 for Sustainable consumption and production.

Our research on conceptual modelling the MSW management system is the first such study in Belarus. The most recent studies focus on the existing approaches to waste management [50], discussing the legislation in hazardous waste management [51] and utilization of industrial waste [52]. The aim of the study was to formulate the model of a MSW management system in Belarus. In order to achieve the above, the following objectives were formulated and solved. Firstly, a descriptive analysis of the current state of the MSW management system in Belarus was conducted. Secondly, the findings have been discussed and suggestions for improving the waste management policy at the municipal level of spatial planning have been provided. The results of the research are presented in this article. The review is organized as follows: Section 2 involves the general state of knowledge in MSW management concerning Belarus. In Section 3 we describe the concept of MSW management including its general principles, hierarchy and methods. In Section 4 we analyze the current status of the MSW management system in Belarus. Section 5 provides the suggestions on formulating the conceptual model of MSW management regarding the specifics of Belarus, including informational, organizational, technological and economic bases as well as the local and regional context. Finally, the conclusions of our study are presented in Section 6.

## 2. MSW Management in Belarus—General State of Knowledge

The amount of MSW collected is increasing annually in Belarus, therefore the environmental, social and economic risks are also remaining here. MSW disposal in Belarus has doubled within the last 15 years from 10 million m$^3$ in 2002 to 22 million m$^3$ in 2018, while MSW composition is getting more complicated, including an increased volume of environmentally hazardous components [53]. Landfills pollute the atmosphere, surface soils, groundwater, negatively affect flora and fauna, and worsen the living standards in nearby settlements along with the land resources withdrawal.



Currently, there is no integrated system of waste management in Belarus—the economic aspect is crucial against social and environmental ones. The MSW recycling rate in Belarus is 18.8% (2018), whereas some countries of the European Union (EU) have a MSW recycling rate exceeding 60% (2018). Therefore, more than 80% of MSW in Belarus goes to landfills. According to NGOs [54], waste management requires structural changes aimed at reducing the environmental impact and increasing resource efficiency.

Since 2007 the Belarusian government has implemented a number of legislative and regulatory documents aimed at ensuring an integrated waste management policy aimed both at reducing the environmental impact and increasing the efficiency of resource and energy use. Unfortunately, the proposed legislative reforms rarely materialise adequately in practice.

The average amount of MSW collected in Belarus is about 4 million tons annually as for 2020. In addition, there are big differences in MSW generation per capita between urban and rural areas. Increasing waste generation is strongly related to increasing public welfare, i.e., there is a correlation between the dynamics of GDP and waste generation per capita.

Being the leader in waste management among the countries of the Eastern Partnership (EaP) region, Belarus retains a high level (402 kg) of waste generation per capita as compared with the EU countries (Figure 1). Accordingly, the rates of municipal waste per capita in EU (2018) were the highest in Denmark (814 kg) and Luxembourg (803 kg) and the lowest in Hungary (381 kg), Poland (329 kg) and Romania (272 kg). Among the EaP countries neighboring EU Moldova generated the highest volume of MSW (776 kg) and Ukraine the lowest (268 kg) [55,56].

It should be noted that of all the efforts towards integrated waste management initiatives, landfilling remains one of the prevalent ways of waste disposal. An average of 22% of MSW in the EU was disposed at landfills in 2018 (there are more than 500,000 landfills in the 27 EU countries). Comparatively, the rate of non-recyclable waste in EaP-countries neighboring the EU is significantly higher. For example, Belarus has a rate of about 80%, Ukraine 96% and Moldova 100% (based on 2018 data, UNSD/UNEP Questionnaire on Environmental Statistics) [57]. In order to create an efficient MSW management system in Belarus it is important to consider and adapt the best EU practices.

Currently, there are only the seven largest cities in Belarus that have modern waste treatment plants and a functioning system of waste segregation and recycling. It should be noted that capacities of waste treatment plants are not sufficient to recycle municipal waste. The lack of a comprehensive approach to MSW management remains a damaging environmental and public health impact. While the situation has been gradually improved over recent years, the MSW recycling rate, even according to official statistical data (about 25% in 2021), is still significantly lower than the EU average (about 40%). Additionally, there are no waste incineration plants in Belarus yet. Thus, most of over 4 millions tons of municipal waste collected annually is transported and deposited in landfills, primarily in rural locations. The existing landfills have a huge negative impact on the rural environment due to inadequate technical equipment. Despite the fact that legal regulations on waste management in Belarus are oriented to EU concepts and are the most efficient among the EaP countries, there is no local and regional coverage at all, particularly for rural settlements. In addition, there is a lack of research on waste management in rural areas in Belarus; therefore, the topic of the research is particularly relevant.

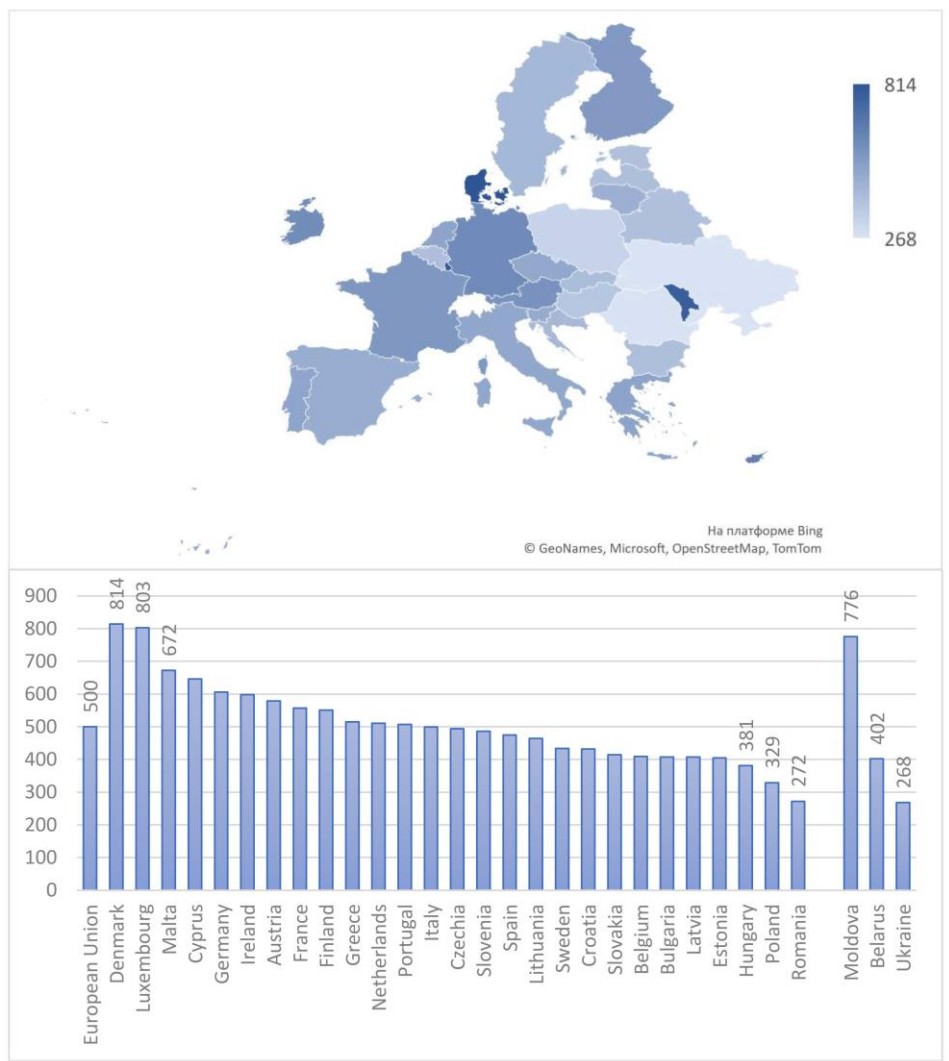

**Figure 1.** Waste Generated in EU and Countries of Eastern Partnership (2018, Kilograms per Capita). Source: Own Study using data from [55,56].

## 3. The Concept of Integrated MSW Management

### 3.1. General Principles of MSW Management

The urgency of the waste problem is related to the global scale of its generation. According to the World Bank, the world generates over 2 billion tons of MSW annually [58]. The necessity to solve the waste problem has caused a new area of environmental policy aimed at the development of methods in MSW management and treatment. The term 'waste management' means the regulating and managing of all processes associated with waste generating, storing, transporting, recycling, treating and disposing of waste, has emerged and become widespread worldwide to describe this area.

The waste issues are to be solved by implementing the integrated system of waste management [59]. The main target of waste management in accordance with the sustainable development paradigm is reduction in landfill waste and maximization of recycling as secondary raw materials and energy sources [60,61]. Considering the above, waste management should follow a comprehensive approach including technical, financial, social, cultural, environmental and governance dimensions [62,63].

There are various interrelated aspects of MSW crisis making (in addition to the waste landfill space shortage): (i) MSW volume is constantly increasing, either in absolute values or per capita; (ii) MSW structure becomes more and more complicated, including an increasing share of environmentally hazardous components; (iii) community reaction

towards waste landfilling techniques becomes extremely negative; (iv) legislation tightening the waste management regulations is implemented at all government levels; (v) innovative waste management technologies, including advanced segregation systems, incinerator plants and sanitary landfills, are widely implemented; (vi) waste management economics becomes more complicated and waste management costs are rising rapidly, thus waste management is impossible without private players and large investors. Both of the above make a 'vicious circle' of waste management crisis as shown in Figure 2.

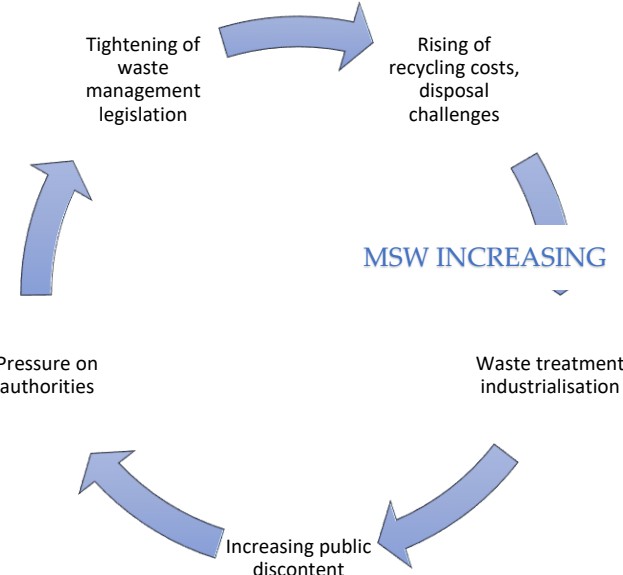

**Figure 2.** The 'vicious circle' of waste management crisis. Source: Own Study.

The current critical situation in MSW management requires new solutions for reduction and minimization of environmental waste impact: a complex of social, economic, technological and engineering problems in waste management caused the necessity of developing the concept of integrated MSW management [64,65]. Such concepts were developed, adopted in numerous world countries and approved within their legislative bodies. Thus, there are two laws About environment protection and About waste management in Belarus, *Natural Resources Conservation Law* in USA, and *Waste Framework Directive* in EU. While the above documents consider the features and level of national development and vary slightly, the basic concept of MSW management remains common regardless of the location or economic position of the country. Therefore, the basic idea of the MSW management system is to minimize MSW disposal or landfill, while the general principles of integrated MSW management systems are multicomponent nature, treatment combination, community outreach, flexibility and adaptability, and stakeholder involvement (Table 1).

*3.2. MSW Management Hierarchy*

Despite an advanced legislative background [66], the theoretical and methodological bases in formulation of integrated system of environmental protection and rational environmental management, including waste management, have not been completely developed in world practice yet [67,68]. Practical approaches to creating such a system include mechanisms and tools for implementation of optimal MSM policy [69,70] formed under the influence of a wide range of external and internal pressures [71,72], and vary significantly from one country to another [73,74].

**Table 1.** General Principles of Integrated MSW Management Systems.

| Principle | Description |
|---|---|
| Multicomponent nature | MSW systems consist of different components, which should be treated with various approaches |
| Treatment combination | All technologies and measures, including waste recycling and backfilling, landfill and incineration should be combined for MSW management by components. Each technology and measure should be developed as a comprehensive and mutually reinforcing package |
| Community outreach | MSW management systems should be based on specified local challenges and resources. Local knowledge on MSW management should be accumulated step by step through the development and implementation of local programs |
| Flexibility and adaptability | An integrated MSW approach is based on strategic long-term planning, providing the flexibility required to adjust for potential changes in MSW composition and volume, and the availability of recycling technologies. Monitoring and evaluating the outcomes should be an integral part of program implementation |
| Stakeholder involvement | Involvement of municipal authorities and community groups—'waste generators'—is a required step in solving the MSW problem |

Globally, the principal routes of waste management were defined at the International Conference on Sustainable Development in Johannesburg (South Africa) in 2002, including waste prevention, maximizing reuse as well as recycling and the use of alternative, environmentally friendly materials.

Generated MSW is treated using the methods which may be conveniently classified into three main groups as follows: (i) recycling as returning of separate components of MSW into economic circulation by separating them from the whole volume and utilizing contents as raw materials and production outputs; (ii) backfilling as utilization of organic components of MSW through biological treatment by various microorganisms; and (iii) incineration as using of mixed MSW or separated fractions to generate thermal and/or electric energy.

In the EU, the legislative framework for waste management is set by two main directives—the *Waste Framework Directive* and the *Hazardous Waste Directive*. The requirements of the directives are implemented via the national legislative systems of EU member states. The hierarchy of waste management methods (in descending order of priority) is currently legislated in the EU by the Waste Directive as follows: prevention; preparing for re-use; recycling; recovery; and disposal. Waste prevention is the preferable strategy, and landfilling should be the last solution. Waste preventing and recycling provide an overall reduction in waste generation [75].

MSW prevention is one of the first and important stages of the waste management system, aimed at reducing the volume of waste at the outset. Waste prevention is achieved by reducing the total amount of waste generated and reducing its toxicity. To reduce the amount of waste, a recycling system is implemented and must be legally enshrined. For example, Germany has a circular economy law requiring producers to achieve zero-waste production and, if it is not possible, the waste must be recycled as a material or energy input. Thus, there is an increased stimulus to use recyclable materials such as cardboard, paper and aluminum foil for packaging.

MSW Re-Use means any action whereby products or components that are not waste are reused within the same purpose that they were intended for.

MSW Recycling has a significant role in integrated MSW management and provides new sources of energy, reduces the use of natural resources and prevents landfilling.

MSW Recovery either involves thermal incineration at waste incineration plants (incinerators) or waste disposal at environmentally secure landfills. Some MSW cannot be recycled via any currently known method (this is called residual waste), and has to be incinerated or disposed at landfills. Incineration plants also provide supplementary energy during the disposal process.

MSW Landfilling is ranked last in the MSW management system, but still remains as a necessary disposal source for non-recyclable wastes. Such wastes comprise incombustible or incinerated components with large toxic emissions that cannot be captured or require unknown or too expensive air emission treatment systems.

Within such an integrated approach towards MSW management, an essential item at any level is segregation as a way of separating hazardous and valuable substances away from MSW.

### 3.3. Methods of MSW Management

MSW management as an integrated item of the overall system of public administration is based on the main interrelated management approaches including organizational, educational, economic, technological, informational and legislative (Table 2).

**Table 2.** Methods of MSW Management.

| Method | Description |
| --- | --- |
| Organizational | Creation of governance, production and monitoring frameworks |
| Educational | A range of measures for achieving the required level of community environmental culture and the professional education of specialists |
| Economic | Encouraging businesses to become greener, to use natural resources rationally, and to reduce the generation of waste |
| Technological | Identifying or establishing instruments and procedures for the environmentally sound production and sustainable waste management |
| Informational | Creation of a waste database, a system of waste accounting, collection and disposal, and the ecological condition of territories |
| Legislative | Preparation of legislation, policy, instructional and regulatory documents defining standards and regulations for waste management |

The basis of any management system is the organizational and legal framework that defines the algorithm of activities in waste management.

The information and educational framework should provide the necessary information comprehensiveness to make well-grounded managerial and production solutions.

One of the most important instruments of waste management is economic regulation methods, in which pollution fees serve as a management lever. Pollution fees are a compensation of economic damage from emissions and discharges of pollutants into the environment, as well as waste disposal on the state territory. To implement an economic motivation in waste management benefits may be provided, for example: preferential payments for waste disposal while using new technologies that provide waste reduction; and establishment of credit benefits for individuals and companies engaged in waste management activities.

The technological framework includes selection of the best and innovative techniques for waste collection, sorting and recycling, which are required for sustainable waste management.

## 4. Analysis of the Current Status of MSW Management System in Belarus

### 4.1. Specifics of MSW Management in Belarus

The general document, which regulates the activity in waste management in Belarus, is the Law of the Republic of Belarus 'About environmental protection', following which the Law of the Republic of Belarus ´About waste management´ was accepted. The necessity of the above documents is caused by the need to control the processes of generation, collection, accumulation, transportation, storage, utilization and disposal of wastes, complying with the Belarusian Constitution, natural environment legislation, the Basel Convention on the control of transboundary movements of hazardous wastes and their disposal, ratified by Belarus in 1999.

The Law of the Republic of Belarus 'About Waste Management' provides the principles of waste management, which define the following ways and variants of waste management in priority ranking: minimization of waste generation by preventing its production, utilization of waste for manufacturing products, energy, works, services; regulated waste neutralization; waste disposal as the lowest priority way of waste management. The state

program ´Comfortable housing and friendly environment´ for 2021–2025 is aimed at further development of housing and communal services (HCS), domestic services, increasing the availability of energy and gas supply in settlements.

The information campaign on responsible waste management called 'Target 99′ has been running in Belarus since 2015. The initiative was created as a unified information campaign to promote responsible citizens' behavior towards waste management, to popularize the recycling and waste segregation, and to encourage people to segregate as much waste as possible. Subsequently, new facilities have been created, recycling capacity has been increased, segregation of mixed municipal waste has been organized in the largest cities, and much machinery and equipment has been purchased. However, all this has not led to significant changes in waste management efficiency.

Today, MSW management in Belarus operates according to the National strategy for the management of MSW and secondary material resources. The specificity of Belarus is the difference between regional territories (from a large city with a high population density to small dispersed and isolated rural settlements), so there should be a regional approach in MSW management system. However, the issue receives limited priority at the state level: only the Strategy of integrated MSW management for Minsk region was developed at the regional level with the financial support of the EU and the involvement of sectoral NGOs, which are currently restricted or closed due to the existing governmental system.

At the same time, Belarus meets the challenges due to membership in post-Soviet unions. For example, in 2020 some EAEU countries did not support the initiative of Belarus to prohibit the use of some types of packaging (PVC labels, lightweight plastic bags with a thickness below 50 microns, and expanded polystyrene packaging for food products) as well as using oxidizable agents in the packaging production. Discussions are still ongoing for now.

Belarus has a single national operator responsible for MSW management. The actual mechanisms of collection of secondary material resources (SMR) among municipal wastes in Belarus include: (i) SMR collection via a system of collection facilities, amounting approximately to 1700 units; (ii) segregated collection of waste from the population by special containers for separate collection of SMR with additional segregation / re-sorting; and (iii) segregation of mixed MSW at the sorting lines, waste recycling and waste sorting facilities.

SMR types for mandatory collection are specified in accordance with the available methods of waste utilization and needs of waste-using industries in the country. There is one way of waste utilization available in Belarus, namely using wastes as secondary raw materials within new production, or so-called material recycling. Currently there are seven waste recycling plants located in regional centers (Brest, Gomel, Grodno, Mogilev, Minsk), subregional centers (Baranovichi and Novopolotsk) and 80 sorting and re-sorting lines in Belarus [76]. Wastes of paper, glass, polymeric materials, used tires, used automobile oil, and electronic wastes are recycled in Belarus. The recycling facilities processing the main MSW types are shown in Figure 3.

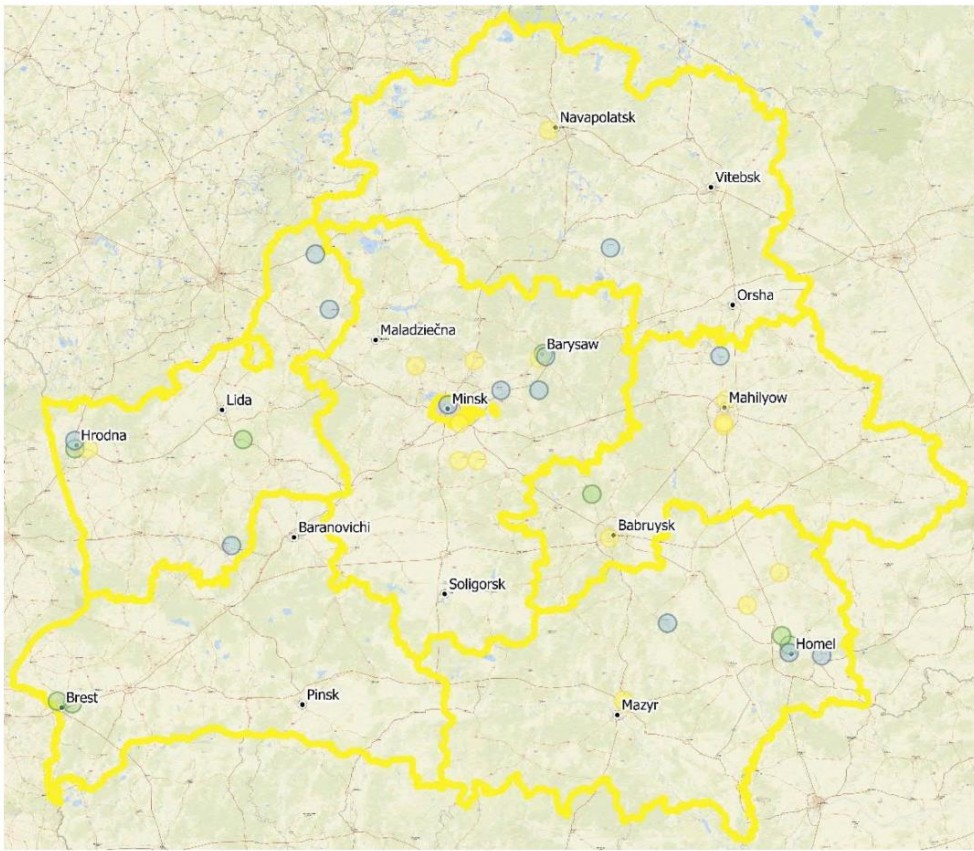

**Legend**:
- 🔵 Paper and carton waste recycling
- 🟢 Glass waste recycling
- 🟡 Plastic waste recycling

**Figure 3.** Recycling facilities for the main SMR types in Belarus. Source: Own Study using data from [77].

Regardless of any measures implemented, the waste recycling situation in Belarus remains troubled. MSW segregation facilities are overloaded: annually over 3 million tons of MSW is generated in Belarus, and this amount increases every year by 20% (according to the official statistics). The sorting capacities (790 thousand tons) considerably exceed the recycling capacities (338.7 thousand tons per year) (Figure 4). Until this imbalance is eliminated, SMR will continue to arrive at landfills as a result of being unclaimed.

There are no incineration plants in Belarus yet. The *Strategy for Waste Management* planned to open a waste incineration plant near Minsk in 2023; however, considering the actual political and economic situation in Belarus it is unlikely that the plant will be ready for use as scheduled.

### 4.2. MSW Management Hierarchy in Belarus

The implementation of MSW management methods varies significantly from country to country depending both on socio-economic development and a number of local factors and country-specific features. Unused MSW is disposed in specialized areas—landfills, considering the environmental protection requirements. Figure 5 shows MSW treatment by type of recovery and disposal in EU and some countries of the Eastern Partnership (EaP).

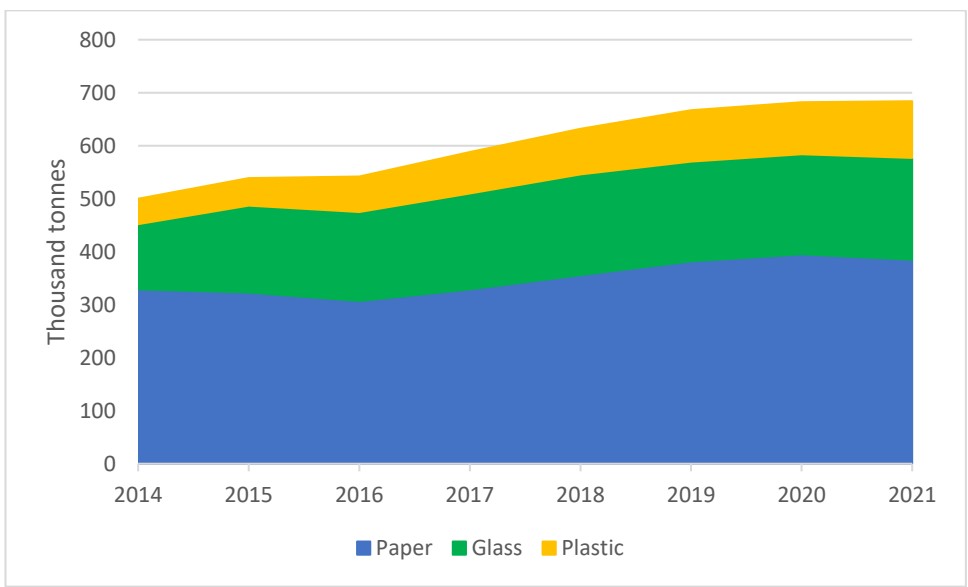

**Figure 4.** Amounts of secondary material resources collection in Belarus. Source: Own Study using data from [76].

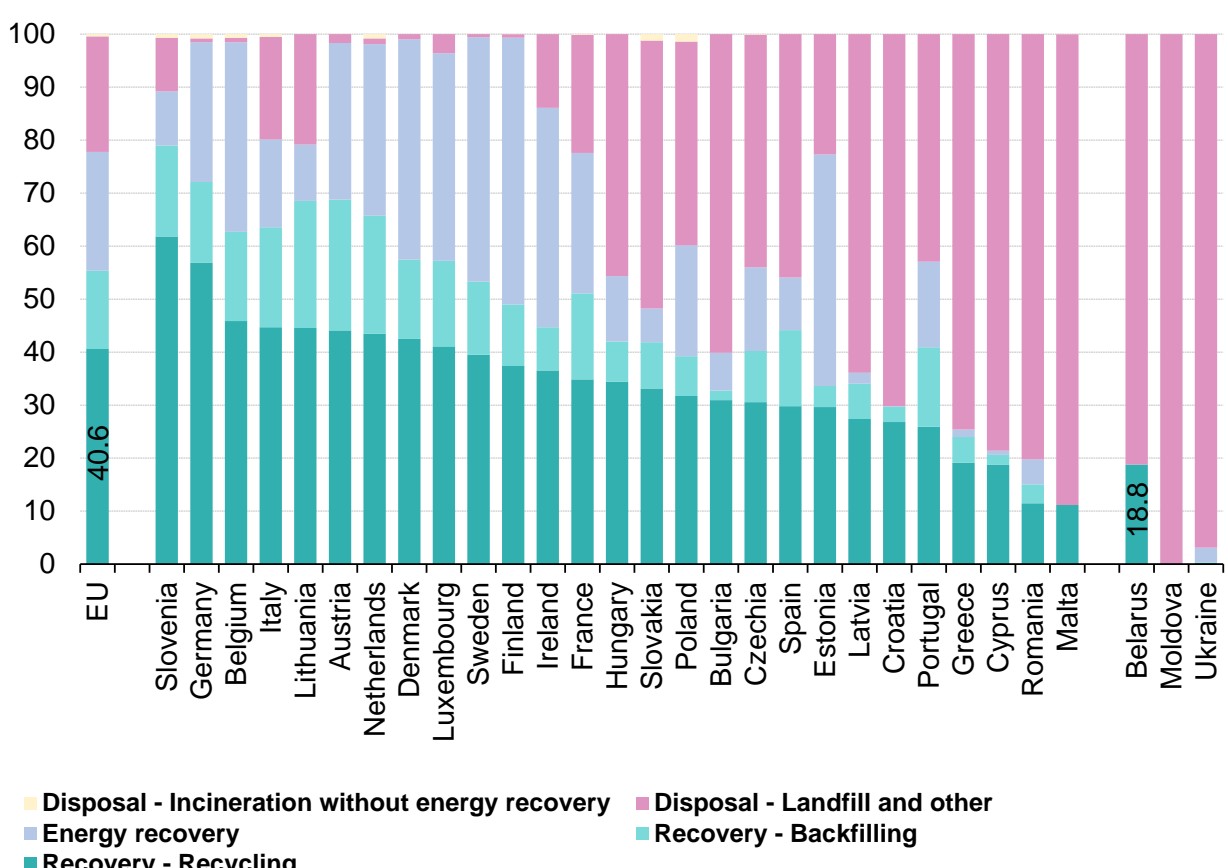

**Figure 5.** Waste treatment by type of recovery and disposal in EU and EaP Countries, 2018 (% of total treatment). Source: Own Study with using data from [57,78].

While in the EaP countries the prevailing MSW method is the landfill, in the EU there are significant regional differences in treatment and disposal methods. The EU member states with the highest share disposed in landfills MSW are Malta (88.9%), Romania (80.2%), Cyprus (78.6%), Greece (74.6%) and Croatia (70.2%). The highest rate of MSW incineration

is in Finland (50.3%), Sweden (46.1%), Estonia (43.7%), Denmark (41.6%), Ireland (41.4%), Luxembourg (39.2%), Belgium (35.8%) and Netherlands (32.4%). Recycling is most common in Slovenia (61.8%), Germany (56.9%), Belgium (45.9%), Italy (44.7%) Lithuania (44.6%), Austria (44.1%), Netherlands (43.4%), Denmark (42.5%) and Luxembourg (41.1%). The Member States with the highest rate of backfilling are Austria (24.7%), Lithuania (24%), Netherlands (22.3%), Italy (18.9%), Slovenia (17.2%) Belgium (16.8%) and Luxembourg (16.2%).

Figure 6 shows the hierarchy of MSW management in Belarus in relation to the structure of the EU system.

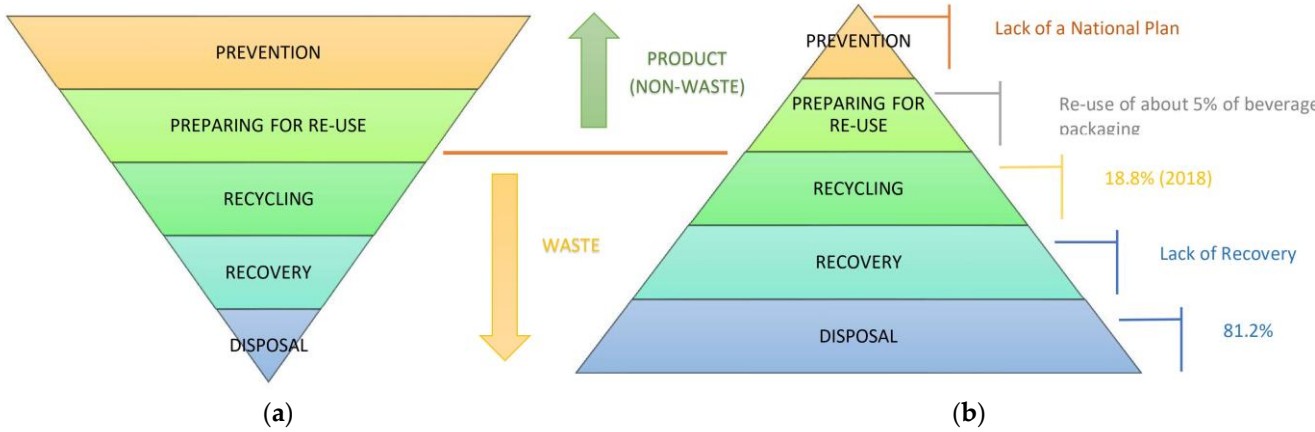

(**a**)　　　　　　　　　　　　　　　　　　　　　　　　　　　　(**b**)

**Figure 6.** MSW management hierarchy: (**a**) according to EU Waste Framework Directive; (**b**) existing in Belarus. Source: Own Study.

### 4.3. Barriers and Challenges to MSW Management in Belarus

Considering the above, we have identified and classified several barriers and challenges related to MSW management in Belarus. The classification was made by four basic aspects in accordance with the waste management model defined in Section 3.3, including (1) organizational and Legislative; (2) economic; (3) technological; (4) informational; and educational (Table 3). It is obvious that the revealed challenges require an integrated approach involving significant financial investment and clearly formulated government programs to support waste management industry as a whole.

**Table 3.** Barriers and challenges to MSW management.

| MSW Management Aspect | Barriers and Challenges |
|---|---|
| Organizational and Legislative | Urban planning policy ignores new requirements for segregated MSW collection (planning of courtyard areas, design of dwellings with waste chutes) |
| | The ban on disposal of recyclable waste resources defines neither the objectives, mechanisms and timeframes for achieving this legislation, nor links it to the phased implementation of an exemption for the disposal of untreated waste |
| | The hierarchy of waste management preferences is incompletely defined at the legislative level |
| | A lack of objectivity in the tendering process among MSW treatment companies |
| | Lack of centralized coordination of regional location of SMR recycling entities |
| | Lack or formality of regional strategies of MSW management |
| | Lack of legislative green requirements to energy usage and biological treatment of the organic fraction of MSW |

**Table 3.** *Cont.*

| MSW Management Aspect | Barriers and Challenges |
|---|---|
| Economic | Poor provision of containers for separate collection of MSW<br>Lack of economic efficiency of the segregated collecting and recycling process due to unjustified fee-setting for MSW management<br>The authority of local governments in the area of MSW and SMR management is not adequately supported by implementing tools, primarily economic ones |
| Technological | Imperfect analytical accounting system for MSW and MSW, including sources of generation, disposal, as well as approaches in determining the morphological structure and generation norms<br>Obsolete facilities and equipment of the existing MSW management system<br>Poor share of specialized providers of MSW collection, removal and disposal services<br>Low recycling of polymer waste—the most environmentally hazardous<br>Inability to ensure safe disposal of MSW in line with current regulations as a result of insufficient upgrading of existing landfills |
| Informational and Educational | Insignificant investments in public outreach<br>Negative community attitudes towards adequate waste segregation and recycling<br>Poor community knowledge on waste management<br>Insufficient community involvement in decision-making due to inefficiencies in the local self-government system and distrust towards current authorities |

## 5. Suggestions on Formulating the MSW Management Model in Belarus

### 5.1. Informational Basis of MSW Management

Currently there is a lack of appropriate informational instruments for MSW management in Belarus. Within the information project called 'Target 99' launched in 2015, an internet portal containing countrywide information on types and volumes of recyclable waste, locations for waste collecting and recycling has been created [76]. One of the possible solutions for improving the information component of MSW management system is to develop an informational model for waste management.

The key informational aspects and tools for improvement of MSW management are shown in Table 4.

**Table 4.** Key aspects and tools for improving MSW management system.

| Aspects | Tools |
|---|---|
| Raising the overall knowledge on the waste impact on the environment and human health<br>Fostering a responsible attitude in using resources and clarifying the benefits of waste sorting and recycling<br>Providing information on the types of MSW suitable for recycling, benefits, specific features and drawbacks of different methods of waste management, and the consequences associated with applying such methods in any certain region or locality<br>Informing about specific aspects of the existing and emerging legal requirements, programs and initiatives, financial resources and compliance procedures | Social media and ambient media<br>Visual information on technologies and methods of waste management<br><br>Training courses and seminars, including educational organizations<br><br>Realization of experimental programs and demo projects |

One of the multi-functional tools is the information model of the waste management system. The main aim of model creation is formulation of an integrated information system with extensible functionality to ensure clarity in waste management systems, and simplify interaction of all stakeholders: waste producers (inhabitants); governmental authorities; companies engaged in waste management; authorized environmental organizations; and other stakeholders. System functionality involves solving a number of objectives, as shown in Table 5. The model represents a GIS database comprising the following elements: basic topographic layer and overhead layers including waste management facilities (landfills, collection stations, separation facilities, etc.); waste sources; operating zones; waste management facilities; garbage routes; container facilities; etc.

**Table 5.** Objectives and functions of an MSW informational model.

| Objective | Function |
|---|---|
| Formation of a GIS-linked database on waste management | Provision of useful and regulatory information |
| Creation of a public information system with the ability of functionality extension | |
| Visualization of information on waste traffic schemes and waste balances | Generation of waste stream balances at regional, municipal and local levels |
| Reduction in time for providing waste management services | Organization of a layered e-monitoring system |
| Stakeholders integration into a centralized information system | Integrated systems of billing and payment for waste management services |
| Feedback between waste operators, government authorities, citizens and other stakeholders | The open component of the model allows for both an interactive map and a fully functional communication portal to residents and organizations. |

Therefore, the key elements of an optimal electronic model include: (1) a database describing the current system status; and (2) an e-portal for stakeholders communication. A complementary component is a variety of operational modules ensuring various services for the participants.

Considering that the waste management sector is developing quite rapidly and is constantly updated with new regulations, the e-model may serve as a flexible tool for all stakeholders of waste management [79,80]. For example, government authorities will be able to react promptly to the situation and introduce amendments to the scheme of waste management in time, to analyze the efficiency of operators' activities. The population becomes more involved and informed as the process of waste management gets clearer and more controlled. It should be noted that such an informational model fits into the most complicated Smart-concepts [81,82].

*5.2. Organizational Basis of MSW Management*

Organization of waste management requires the participation of all stakeholders (local government, business community and rural community). The process flow of a MSW management organization includes 4 main stages as shown in Figure 7.

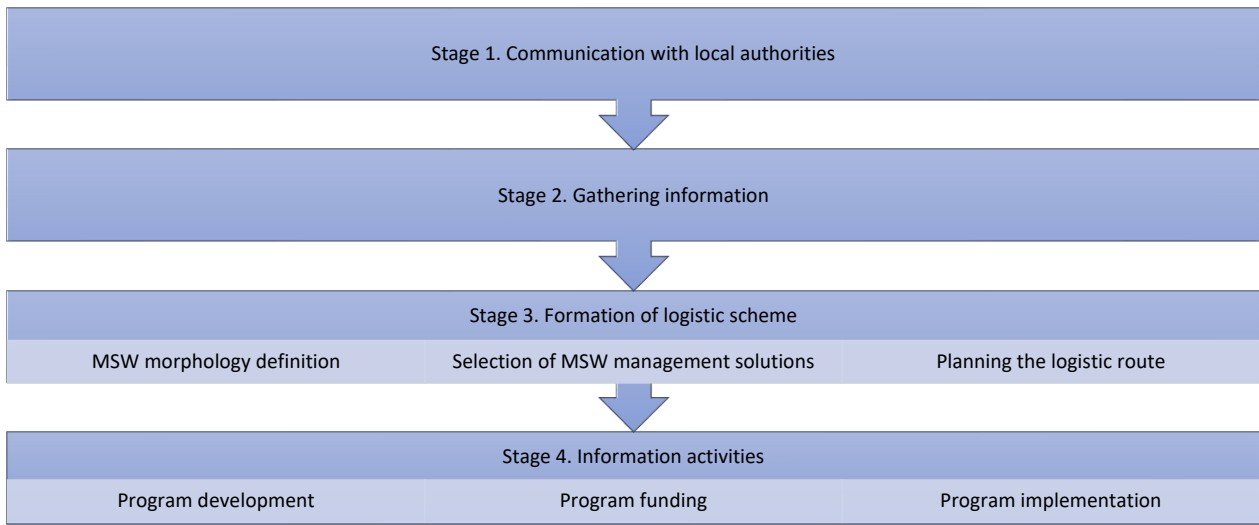

**Figure 7.** The process flow of MSW management.

First stage. Communication with local authorities to provide community initiatives and business proposals.

Second stage. Collection of information on relevant legal acts, regulating the issues of waste management at the local level of land-use planning.

Third stage. Formation of a logistic scheme, including definition of waste morphology in order to estimate fractions, selection of waste management techniques, planning the logistic route in accordance with the initial data.

Fourth stage. Information activities. The process of informational education on waste management involves the following steps: formation of a workgroup; development of an informational program (lectures, seminars, festivals, videos, handouts, etc.); funding; and program implementation.

MSW management plan according to Waste Framework Directive is shown in Table 6.

**Table 6.** MSW management plan.

| MSW Management Level | Suggestions |
|---|---|
| Prevention | Extending the life of consumer products<br>Centralized collection of organic waste<br>Educational and training programs |
| Re-Use | Separate collection of food waste<br>Collection of recyclable packaging at the stores<br>Formation of recycling collection centers<br>Economic motivation for separated collection and reuse |
| Recycling | Utilizing as raw materials<br>Creation of facilities for segregation of non-food waste<br>Heat treatment |
| Recovery | Implementation of composting facilities<br>Implementation of waste incineration facilities |
| Disposal | Implementation of advanced landfills<br>Incineration without energy |

### 5.3. Technological Basis of MSW Management

As the main subsystems in a technological model of MSW management system, we can consider the following: (1) MSW collection; (2) MSW transportation; (3) MSW segregation; (4) MSW recycling; (5) MSW disposal. It seems that the most significant stages in waste management in achieving the aim of zero waste are collection and treatment; therefore, we should discuss them in detail.

There are various relevant schemes of MSW collection in the EU concerning rural settlements [83,84]: (1) Household waste collection involves all variants of MSW collection (in bags, special bags, containers and bins) directly near inhabitants' households. Mixed household waste collection involves partial separation into fractions, for example, plastic and metal are collected in the same container. Separated household waste collection means separate waste collection by fractions, usually in special bags [85]. (2) MSW collection facilities: containers for different waste types are installed in public areas. (3) Waste treatment centers: generally fenced and often staffed collection centers that collect household waste for recycling. Such centers often also receive non-household waste such as hazardous and oversized waste, electronic waste, and construction waste [86]. (4) Package collection stations (fandomats, depository system) are generally used for collecting glass and plastic bottles or metal beverage cans.

The depository system has existed in Belarus since the Soviet period, while its upgraded model has been effectively implemented all over the industrialized world. Currently the successful example is Lithuania, as its population returns more than 95% of beverage containers with a voucher that can be cashed in or used for buying goods. Implementation of the depository system in Belarus, according to the *National strategy of waste management*, was expected in 2020–2021; however, the program was delayed later until 2024. In addition, the approaches towards recyclable waste collection in Belarus differ considerably when compared to the EU experience. A significant volume of commercially valuable waste is still collected via the collection system retained since Soviet times, with administrative regulation of SMR collection as a peculiarity.

It should be noted that the technological model of MSW management is specified by a wide range of determinants including planning, environmental and other aspects [87–89]. For a certain rural community, we should consider the distance between the community

and the treatment facility, type and density of housing, planning concepts, as well as waste management strategy and weather conditions. Selection of the optimal treatment method depends on requirements for environmental protection and public health care [90,91] as well as economic efficiency, environmental feasibility and rational use of land resources [92].

Considering the mentioned aspects and global experience, we have identified the most relevant ways of waste treatment in rural communities: (1) Waste transfer station is an additional facility required for waste transportation to another facility to be utilized or recycled. A fee might be charged depending on the location. (2) Private landfill is a way of disposing of MSW unhazardous by burying on private land, unless prohibited by law. (3) Composting is a way to dispose of organic waste. (4) Waste incineration in the countryside should be legally permitted as well as technically and environmentally safely, followed by ash utilization. (5) Waste collection service provides the safest integrated way of waste management in rural areas. The service fee depends on the selected waste collection scheme. Benefits and drawbacks are shown in Table 7.

**Table 7.** Ways of waste treatment in rural communities.

| Treatment Method | Benefits | Drawbacks |
|---|---|---|
| Waste Transfer Station | Reduction in transport costs<br>Schedule flexibility | Difficulties for residents with waste transportation<br>Environmental impact |
| Private Landfill | No transport costs<br>Schedule flexibility | Hazardous wastes require additional treatment<br>Environmental impact |
| Composting | Extra source of organic fertiliser<br>Decreased volume of organic waste | Unpleasant smells discomfort<br>Extra charges due to the composting facility |
| Incineration | Decreased volume of disposal waste<br>Extra energy generation capacity | Smoke and ash environmental pollution<br>Increased requirement for ash disposal |
| Waste Collection Services | Environmental friendliness<br>No environmental hazards inside community | Additional transport costs for isolated communities<br>Schedule related dependency |

Considering the ways of MSW treatment and organizational suggestions we have formulated the scheme of MSW management as applied to rural settlements in Belarus (the existing and suggested), as shown in Figure 8.

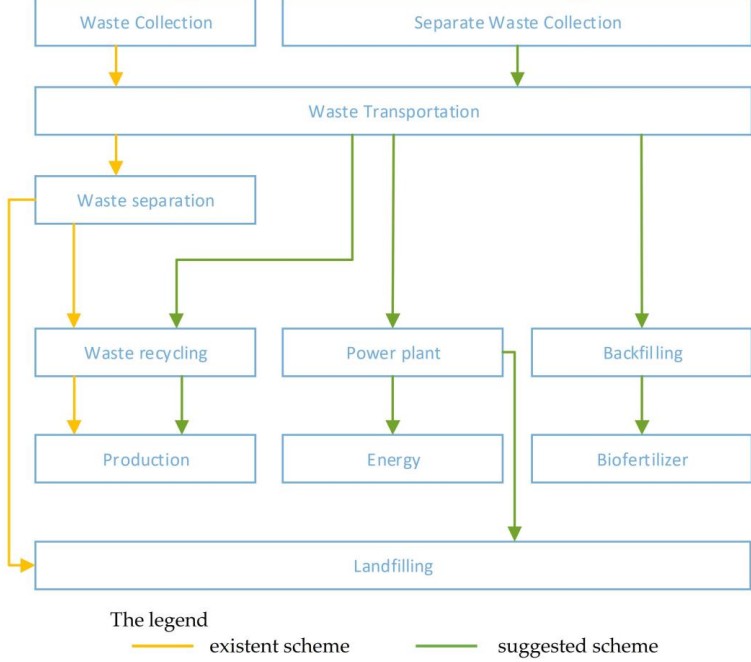

**Figure 8.** The flowchart of waste management in rural communities. Source: Own Study.

### 5.4. Economic Basis of MSW Management

The economic responsibility for waste management is imposed on waste generators, including community inhabitants [93–95]. The rates for MSW utilization in Belarus are economically unjustified instead of being set by administrative regulation. As a comparison, the average monthly rate for MSW treatment per capita in Poland is EUR 6.25, and in Germany EUR 17.5. The correlation of MSW treatment costs with the average monthly salary are as follows: Germany—0.42%, Poland—0.49%, Belarus—0.12% (Table 8).

**Table 8.** Correlation of MSW treatment costs and average salary (2022).

| Indicator | Germany | Poland | Belarus |
|---|---|---|---|
| Average salary, EUR (Euro) | 4091 | 1284 | 661 |
| Monthly average payment for MSW treatment per resident, EUR | 17.5 | 6.25 | 0.82 |
| Share of MSW treatment charges in salary, % | 0.42 | 0.49 | 0.12 |

The internationally approved practice suggests the affordability limit on waste management services per households is 1% of an average income, so it is evident that the rates in Belarus are undervalued. In order to improve the payment system for MSW management the EU experience is useful, in particular the Polish experience as it has a diversified approach to payment for MSW utilization. On the one hand the rates are set at the local level (gmina), but also regulated legislatively at the national level by *Law on maintaining cleanliness and order in local communities* dated September 13, 1996. The primary methods of calculating rates for MSW utilization and average tariffs in Poland are shown in Table 9.

**Table 9.** Billing approaches for MSW utilization.

| Billing Determinant | Average Rate, EUR | Share of Monthly Average Income,% |
|---|---|---|
| Number of residents, person | 8.8 | 2.0 |
| Water discharge, cubic meter | 3.1 (max 34.2) | 0.7 (max 7.8) |
| Living space, square meter | 0.35 | 0.08 |
| Fixed tariff per container/bag (120 l) | 5.7 | 1.3 |
| Fixed tariff per apartment/private household | 18/22.7 | 0.41/0.52 |

### 5.5. Regional Approach towards MSW Management in Belarus

MSW management is a component of settlement development model as an integrated socio–natural–technogenic system, therefore it is based on a number of spatial planning aspects including social, economic, environmental, technological and managerial [96,97]. Additionally, the regional aspect of sustainable development is of great importance in MSW management. Therefore, MSW management based on a regional approach, i.e., development and implementation of MSW management programmes specific to each region followed by further integration within the formulation of national policy of MSW management. It is necessary to provide ecological, economic and social consequences of the decision-making process at regional level, considering the following: no economic activity cannot be justified if the benefits do not exceed environmental damage; environmental damage has to be minimal, as far it provides economic and social state of the region.

The major development goal of a regional MSW management system is the comprehensive employment of all management and resource-saving instruments (environmental, technical, economic, regulative, legislative and informational), which identify the key directions: organisation of a step-by-step system of waste transfer; arrangement of maximal waste recycling and re-use; ensuring an eco-friendly disposal at landfills for non-recyclable waste; implementation of secondary resources market and recyclable materials manufactures; implementing the monitoring of MSW collection, transportation, neutralisation and disposal system at the municipal level.

The specific aspect of rural areas in Belarus is the relative distribution of settlements along with the unequal population density in them. Consequently, the local specifics are to

be considered when formulating the waste management strategy, including the rationale for waste management scenarios and waste infrastructural facilities allocation.

Considering the above, a cluster approach has been suggested for grouping rural settlements and then defining waste management scenarios for them. The classification is based on a differentiated approach towards the provision of social, engineering and transport infrastructures and improvement in rural settlements, with regard to the rational use of local resource potential. The proposed categorization includes the following planning aspects: position in the planning structure (transport infrastructure system); administrative and industrial value and economic focus; demographic capacity; recreational resources and historic-cultural heritage; availability of transport and utility infrastructure. We suggest a three-level classification of rural settlement system: core rural settlements; basic rural settlements; and small rural settlements.

The 1st settlement type forms a central core of the rural settlement system and include agro-towns, village council centres, centres and industrial units of agricultural enterprises. Most of the management and administrative services, social and engineering facilities and economic entities are concentrated here. Such settlements have well-developed functions of welfare services for the inhabitants of all neighbouring areas due to the location within the influence of national, regional and local planning axes. The influence area of the 1st type settlements covers a distance between 3–4 and 15–20 km.

The 2nd settlement type is represented by basic settlements that have sustainable economic and social ties with the first type of settlements, due to the location within the influence of regional and local planning axes. It should be noted that depopulation has been registered concerning such settlements within recent decades.

The 3rd settlement type is characterized by low population or a lack of permanent inhabitants. Settlements are located within the influence of local planning axes and possess historical–cultural, natural or industrial capacity.

Table 10 shows the characteristics of the above types of rural settlements and proposed technological and economic solutions in MSW management.

**Table 10.** Characteristics, technological and economic solutions for MSW management for rural communities in Belarus.

| Settlement Type | Settlement Characteristics | Technological Solutions for MSW Management | Economic Solutions for MSW Management |
| --- | --- | --- | --- |
| Core rural settlements (agro-towns, village council centres, centres and industrial units of agricultural enterprises) | Population size—over 200 people Central core of rural settlement system Concentration of management and administrative services, social and engineering facilities and economic entities Well-developed functions of welfare services for all neighbouring areas Location within the influence of national, regional and local planning axes The influence area radius is between 3–4 and 15–20 km | Integrated MSW management: Segregated MSW collection Recycling centres Centralized composing plant Sorting station Transfer stations Incineration with energy recovery Sanitary landfills | Differentiated payment system for separate and non-separate collection by the number of inhabitants Local facilities for waste collection within the communities |
| Basic rural settlements (villages) | Population size—between 50 and 200 people Sustainable economic and social ties with the 1st type settlements Location within the influence of regional and local planning axes Depopulation within recent decades | Traditional MSW management Segregated MSW collection Transfer stations Home composting | Differentiated payment system for separate and non-separate collection by the number of inhabitants Local facilities for waste collection within the communities |
| Small rural settlements (villages) | Population size—less than 50 people Low or a lack of permanent population Location within the influence of local planning axes Historical–cultural, natural or industrial capacity | Traditional MSW management Segregated MSW collection Household reuse and recovery Home composting | Container or bag rates MSW collection as defined in schedule |

Creation of waste recycling complexes makes economic sense as long as the volume of generated waste is significant, i.e., within densely populated areas. Small rural settlements

remotely located from waste infrastructure are generally unprofitable in waste management. Traditional waste management methods are hampered by a lack of extensive transport infrastructure and insufficient funding from the local government. An essential logistics problem for rural areas involves transferring the waste from the generator (village) to waste disposal facilities. Decreased transport costs are achievable by reducing waste generation, particularly in settlements that cannot easily and economically achieve conventional methods of waste collection.

Rural inhabitants also need to be motivated towards innovative ways of waste management, including segregated waste collection, household composting, etc. For this purpose, information and economic mechanisms should be applied, e.g., public education on environmental responsibility, economic motivation through discounts on separate collection rates, financial support via subsidising the costs of composting facilities, etc. However, it is not only waste management that is forcing innovation—other areas of life too [79,98].

## 6. Conclusions

This review contains formulation of the waste management concept, analysis of the current state of the waste management system in Belarus, and recommendations on modeling the waste management system considering the local planning level in Belarus. The following conclusions were obtained as a result of the research:

The solution of the issue on sustainable waste management in Belarus is associated with a number of challenges, including organizational, economic, technical and informational ones.

The existing system of waste management does not move the country towards the implementation of a circular economy; namely, the MSW management on the basis of sorting and recycling is not encouraged, but on the contrary, the preservation of actual methods of waste disposal, including in rural areas, is stimulated. Waste prevention is replaced by MSW segregation and treatment, which does not meet the actual national strategy.

There is no communication mechanism between stakeholders and decision makers in the waste management sector, i.e., the population, operators and local authorities practically have no levers to solve particular challenges. Sustainable waste management requires a local partnership: such an approach is applied in numerous EU countries, including Poland where an open dialogue and feedback is an effective tool in establishing a waste management model, adapted to the regional specificities.

Another significant aspect is the economic rationale of MSW treatment fees for inhabitants as a way to encourage minimization of generation and separate collection, where effective segregation at origin provides an additional waste value as an SMR.

In addition to structural system drawbacks, one of the challenges is a population distrust towards the authorities, even the local government, to respect the public rights and to implement the suggestions.

Therefore, the challenges of waste management are impossible to solve only by separate measures: a comprehensive approach is required in order to support waste recycling into a manufacturing resource within circular economy. There are two primary approaches for solving the problem of waste management in rural communities. The economic approach implies the minimization of waste generation and should be encouraged both at the level of manufacturers; for example, introduction of fees / taxes on certain packaging, as well as consumers; for example, the establishment of a differentiated system of payment for waste disposal. The technological approach comprises the development of mini-solutions for waste recycling at the places of waste generation, e.g., organics, including the support of similar technologies by producers within the Extended Producer Responsibility (EPR) program.

A number of principal limitations for the research should be mentioned. Currently, a major barrier towards realizing a circular economy in Belarus is the inadequacy of the overall political system, as well as the formal nature of a local government system

that excludes the complete involvement of stakeholders, including inhabitants, private business community and non-governmental organizations in decision-making processes. Consequently, it is impossible to ensure sustainable development without renewal of the political system. Further research directions are related to the formulation of the methodology on assessment of waste management efficiency at the municipal planning level.

**Author Contributions:** Conceptualization, A.H. and S.K.; methodology, A.H. and S.K.; software, A.H.; validation, S.K.; formal analysis, A.H. and S.K.; investigation, A.H.; resources, A.H.; data curation, S.K.; writing—original draft preparation, A.H.; writing—review and editing, S.K.; visualization, A.H.; supervision, S.K.; project administration, S.K.; funding acquisition, S.K. All authors have read and agreed to the published version of the manuscript.

**Funding:** This research received no external funding.

**Institutional Review Board Statement:** Not applicable.

**Informed Consent Statement:** Not applicable.

**Data Availability Statement:** Not applicable.

**Acknowledgments:** The authors are very grateful to the academic editors and reviewers for constructive comments and suggestions, thanks to whom the article received its final shape.

**Conflicts of Interest:** The authors declare no conflict of interest.

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
