# Peer review of "MSW Management to Zero Waste: Challenges and Perspectives in Belarus"

_sustainability, doi:10.3390/su15032012_

Round 1

Reviewer 1 Report

The manuscript is titled as Modelling the MSW Management System in Balearus.I presumed there will be mathematical equations, and based on that some conclusions will be derived from them. However; the article is more like a review, there is no clear mathematical formulation, nor modelling based results.

The authors state: "The research methods were based on the complex and interdisciplinary approaches, including the system approach, factor analysis and comparative analysis." However; the the word factor analysis and comparative analysis can only be found in the abstract.

The manuscript contains extensive literature information and data, but is not clear how they are contributed to the model. I think either the title should be changed, or the detailed model formulation should be added.

Some additional comment:

Please avoid lumped references. Every cited article should be assessed with at least halg a sentence.

Page 2 Line 56: MWS management => should be MSW

Page 8 Line 280 What is SMR means?

Reviewer 2 Report

This manuscript provides comprehensive introduction to basic concepts on waste management from various aspects, but it seems to be more than a review rather than a research work. At the first sight of the title, I wonder that what model the authors have constructed, that what methods have been applied and then that what quantitative results could been obtained. However, after reading this manuscript, I just realized the current states of waste management and the strategies to alleviate corresponding problems. Thus, this work could be accepted only after major revisions.

1.       First, the Abstract of this work does not show the key contributions of this work. Especially, the key quantitative results are not found in this abstract.

2.       ‘zero waste’ in keyword only appears two times in context. Is it appropriate to a keyword?

3.       The interdisciplinary approach shown in abstract, such as the system approach, factor analysis and comparative analysis, are not mentioned in the context. In my opinion, such terms in abstract are too ambiguous. And it is quite hard for readers to link these methods with corresponding findings.

4.       There are more than 3 places where authors cite at least 5 references, which should be avoided.

5.       Line 52, what approach is referred to ‘This approach’?

6.       Line 57, the passive tense is unnecessary.

7.       1.1 part of Introduction gives the basic description of current state of waste management in Belarus and is placed at the end of introduction. In my opinion, such structure should be reorganized so that readers could understand the research gap, research aim and research methods of this work.

8.       Line 341, the number 3.2 in subtitle should be 3.3.

9.       Figure 7 just shows the process flow of waste management, and thus the term ‘Algorithm’ in title is not suitable.

Round 2

Reviewer 2 Report

Accept

Author Response

Thank you for accepting.